# Dried Blood Spot in Laboratory: Directions and Prospects

**DOI:** 10.3390/diagnostics10040248

**Published:** 2020-04-23

**Authors:** Kristina Malsagova, Artur Kopylov, Alexander Stepanov, Tatyana Butkova, Alexander Izotov, Anna Kaysheva

**Affiliations:** Institute of Biomedical Chemistry, Biobanking Group, Moscow 109028, Russia; a.t.kopylov@gmail.com (A.K.); aleks.a.stepanov@gmail.com (A.S.); t.butkova@gmail.com (T.B.); farmsale@yandex.ru (A.I.); kaysheva1@gmail.com (A.K.)

**Keywords:** dried blood spots (DBS), serodiagnosis, ELISA, MS, postgenomic technologies

## Abstract

Over the past few years, dried blood spot (DBS) technology has become a convenient tool in both qualitative and quantitative biological analysis. DBS technology consists of a membrane carrier (MC) on the surface of which a biomaterial sample becomes absorbed. Modern analytical, immunological or genomic methods can be employed for analysis after drying the sample. DBS has been described as the most appropriate method for biomaterial sampling due to specific associated inherent advantages, including the small volumes of biomaterials required, the absence of a need for special conditions for samples’ storage and transportation, improved stability of analytes and reduced risk of infection resulting from contaminated samples. This review illustrates information on the current state of DBS technology, which can be useful and helpful for biomedical researchers. The prospects of using this technology to assess the metabolomic profile, assessment, diagnosis of communicable diseases are demonstrated.

## 1. Introduction

The idea envisaged by Ivar Christian Bang (1869–1918) was pivotal to the discovery of cellulose paper cards for the collection of blood samples as a membrane carrier (MC). Initially, Bang quantified glucose concentration from dried blood spot (DBS) components and, later, also performed nitrogen measurements using the Kjeldahl method with this filter paper technique. Since then, many researchers have used his technique to analyze serological samples [1]. The development and improvement of traditional laboratory immunohistochemistry, immunochromatographic assays and mass spectrometric methods for the detection of biological molecules has become increasingly important in biomedical research. One of the primary challenges facing modern laboratories is the demand for novel approaches capable of detection at a molecular level. Thus, molecular profiling of the mechanisms associated with physiological processes and biological diversity remains unsolved in the field of fundamental biomedical research problems related to the inventorying of molecular components in biological samples (the Human Proteome Project, the Immune Proteomics Project, the Animal Toxin Annotation Project, etc.). Furthermore, in the field of applied problems in biomedical research, the most prevalent issue involves the identification of serological protein markers associated with the development of common pathologies with highest level of morbidity and mortality, including cancers, diabetes, and cardiovascular diseases [2,3,4,5]. Blood plasma serves as an attractive source of candidate protein markers and specific pathologies for molecular profiling, as it contains molecular components secreted by cells in diseased tissues, as well as factors involved in the development of pathophysiological processes [6]. Currently, the primary concept vectors described in the P4-medicine strategy (predictive, preventive, personalized, and participatory) include the detection of pathology-specific molecular profiles or disease “signatures” [7].

Due to the ability to establish a comprehensive view of multimodal disturbances in biological processes, “omics” technologies are of growing interest in identifying the molecular patterns associated with socially weighty diseases. “Omics” technologies implement a holistic view of the molecules that make up a cell, tissue and organism. They are designed to detect genes (genomics), mRNA (transcriptomics), proteins (proteomics) and metabolites (metabolomics) in a specific biological sample, in a non-targeted and non-biased manner [8].

Moreover, the current state of society and recent advances in medical technologies have created an urgent need for more balanced organizational solutions for diagnostic laboratory techniques [9]. Hence, the standardization of guidelines for the collection, delivery and storage of biomaterials from patients ensures optimal laboratory operation and, ultimately, accurate etiological diagnoses. Alternatively, there are plenty of common errors within the pre-analytical stage, including non-compliance with biomaterial collection guidelines, violation of storage and delivery conditions and insufficient annotation of biological samples. Violation of these rules leads to incorrect results during the analytical stage, thereby generating a myriad of unnecessary information that significantly diminishes reproducibility [10]. Furthermore, errors in the post-analytical stage, including standardization of data interpretation and statistical analysis strategies, as well as samples from outsiders, also significantly impact on the scientific community. In fact, only 23% of studies successfully confirmed their discoveries via independent verification, as has been concluded as a result of analysis of proteomic research in the field of biomedicine throughout the current millennium [11].

Analysis of the literature reveals that DBS technology is an effective method for the pre-analytic stages of diagnostics [12,13]. However, the development of associated technical methodologies and standardization of procedures for the pre-analytical diagnostic stage has only occurred over the past five years. DBS membrane carriers allow for the transportation of samples to diagnostic laboratories for subsequent serological analysis and long-term storage of biological samples and for generation of biobanks [14]. From the clinical practice view, it is important to note that capillary blood provides sufficient sample volume for processing by DBS. Hence, the required small sample volume reduces the need for the traditional invasive methods of sample collection. Moreover, a biomaterial sample can be delivered to analytical laboratories in packages with variable isothermicity (from +5 °C to −35 °C). It is important that the biomaterial on membrane carriers in dried spots exhibits resilience to long-term storage and does not require special storage conditions compared to cell or blood plasma samples [15]. Therefore, the DBS technique has been an invaluable step in the field of clinical laboratory diagnostics during the past 100 years. It is frustrating that, at a current level of blood sampling and collection for a variety of clinical purposes, the pre-analytical phase continues to be largely undervalued in many other fields in which DBS testing is applied [1].

The use of DBS in clinical practice is preferable to whole-blood sampling, since drying of blood biomaterial reduces the risk of contamination with infectious and other pathological bioagents [16]. It is worth noting that the DBS specimen is a “universal tube”, providing the ability to analyze, in a single sample, a wide range of biomarkers from cells (CD4-T) [17], viruses, nucleic acids (DNA) [10], RNA [18], antibodies (against HIV-1 [19], IgА [20]) and antigens (p24) [21].

Furthermore, the current procedures for DBS are well-standardized quality control methods that permit effective analysis of population studies [20], monitoring of antiretroviral therapy [19], as well as the diagnosis and identification of genetic characteristics associated with HIV-1 infection using PCR [18].

Hence, MC technology not only offers a convenient material for transporting blood samples, but also serves as a key component in multidisciplinary research.

## 2. Search Strategy

To address these aims, relevant studies were identified by a literature search in PubMed, Scopus and Web of Science with the following terms: (“DBS” OR “dry blood spot” OR “filter paper” OR “dried blood filter” OR “dried blood” OR “dried sample”) AND (“MS/MS” OR “mass spectrometry” OR “ELISA” OR “PCR” OR “qPCR”) AND (“screening” OR “serodiagnosis” OR “biobank” OR “biobanking”). In addition, databases were restricted to English language and human species. Restriction to the period of publication was not applied. Selection of eligible articles was built on the careful screening of titles and abstracts. As a result of this selection criteria, 108 references were identified for review. For presentation, the collected information was divided into sections of analytical methods and practical applications. It is not our intention in this review to highlight all analytical and practical aspects pertaining to the DBS approaches but rather to address the several aspects pertinent to the combination of DBS and standard molecular identification methods for biomedical applications.

## 3. Types of Membrane Carriers, Biomaterial Collection, Processing, Storage and Logistics

DBSs collected on filter paper have significantly improved the collection of blood samples in resource-limited settings. There are two general types of DBS carriers in modern research: fiberglass (LLC “Immunologist”, Moscow) and pulp and paper (“Watman”, UK; “Ahlstrom”, USA, etc.). Companies offer different types and modifications of membrane carriers depending on the goals and objectives of the study. For example, the membrane material “Whatman 903” is a Class II medical device approved by the FDA and is often used for biomaterial collection and subsequent analysis of HIV infection [22,23]. Moreover, it was reported as the only membrane carrier used for genotyping HIV variants [24]. Researchers at the Centers for Disease Control and Prevention compared Whatman 903, Ahlstrom 226 and Munktell-TFN membrane materials for quantifying HIV viral load and genotyping drug resistance. The results showed that samples collected with Munktell-TFN carriers exhibited the highest genotyping efficiency (100%) compared to Whatman 903 and Ahlstrom 226 membrane carriers (91.7 %). Moreover, Munktell-TFN was more sensitive for the identification of HIV drug resistance mutations [23]. However, regardless of the DBS chosen for sampling, the general principle of obtaining biomaterial in dry form consistently involves the impregnation of a MC fiberglass strip with blood by immersing its tip in the test liquid, or, if it is blood, transferring a drop of blood from the heel, finger or foot of the patient to pre-printed areas on MC, followed by air drying. Notably, it is not recommended that DBS technology to be used for the analysis of volatile substances or for contaminated or hemolyzed samples [25].

While drying blood samples, it is generally recommended to incubate samples approximately for 2–3 h on an open space at ambient temperature (15–22 °C). It is very important to dry biological samples completely before storage or transportation. Typically, the drying time of biomaterial depends on sample volume and type of membrane carrier [26]. Due care must be taken to ensure that samples are not heated, stacked or allowed to come into contact with other samples or surfaces. Moreover, DBS specimens should be stored away from direct sunlight and high humidity, since both may affect the quality of the biological samples, cause bacterial growth, change the extraction efficiency during analysis or contribute to the degradation of unstable analytes. Therefore, it is recommended to use bags with dehumidifiers for the long-term storage of DBS samples at ambient temperature [27]. Moreover, samples containing unstable compounds should be stored at a low temperature, for example, 2–8 °C, below −15 °C or even less than −60 °C. Professor Cassol has demonstrated that it is possible to store genetic material in the form of a DBS at high temperatures in tropical conditions with no significant losses of sample quality [28]. Specifically, HIV DNA and RNA have been detected after 5–15 years of storage without implementing special cooling techniques [10]. However, in 15-year-old samples, stored at these high temperatures, a certain level of large DNA loci was lost, which was not the case during their storage at −20 °C [29].

A requisite condition for the use of an MC is the ability to effectively absorb biological fluid, evenly distribute it on the work surface and ensure the safety of dried biological material [30]. The presence of roughness and/or porous sublayer significantly alters the wettability of the MC [31] and, consequently, the even distribution of the biomaterial on the surface of the carrier. Hence, selected membrane material should exhibit pronounced hydrophilic properties and a homogeneous structure that allows for uniform and reversible absorption of blood components, including proteins and other biologically active compounds in the pores of the membrane. Surface treatments are, therefore, applied to DBS to improve these properties. One such method, employed for the modification of existing membrane materials in DBS technology, involves treating the surface with nanoparticle and water-soluble polymers, thereby improving their stability and even prompting antibacterial properties. Moreover, treatment of membrane materials with zinc oxide provides improved membrane stability during storage by preservation of the absorbed sample [32]. Further, the coatings with antibacterial properties prevent degradation of biological samples during transport and storage, which might be caused by bacterial contaminants present in the environment (packaging materials, air) during sampling [33]. Lastly, treating surface membranes with hydrophilic glass fibers provides not only the antibacterial properties necessary for stability of the biological material during transport and storage, but also increases the hydrophilicity of the fiber surface [32].

The advantages of DBS are well-known: a small volume of biomaterial (10–40 μL) and low storage and transportation costs. However, several principal factors limit the use of DBS for analyte quantification [34]. It was shown that hematocrit, blood viscosity, the nature of the analyte, type of MC, the effect of chromatography and determination conditions (temperature, ambient humidity) can lead to a change in the size of the spots and uneven distribution of analytes [35,36]. Thus, punching and extracting a fixed spot size can sometimes entail an unacceptable bias in the analysis results [35,36].

This is one of the reasons why the MC used for DBS technology must meet strict quality requirements: uniformity, the absence of chemical leaching, and minimal chromatography effect [37]. It is quite difficult to evaluate the exact volume of blood applied to the MC outside the laboratory environment. Therefore, in some studies, the blood volume was calculated taking into account the applied blood volume, the area of the perforated disk and the average diameter of the MC. A perforated disc MC with a diameter of 3 mm may contain about 4.3 μL of blood, depending on the level of hematocrit and type of MC [37]. Data from other studies showed that, generally, an MC of 3 mm absorbs 1.46 μL of serum with a standard deviation of 0.05 μL [38]. However, despite the use of the same protocol for collecting biomaterial according to the DBS technology, different hematocrit levels and the initial volume of the bloodstain lead to a substantially varying volume of collected blood and serum. The Clinical and Laboratory Standards Institute Approved Standard NBS01-A6 states that the blood prints on DBS cards for newborn screening have an internal diameter of 12–13 mm and states that 75 μL of blood will fill the selected area, whereas 100 μL will fill the area beyond the contour line [39].

One way to overcome the challenge of spot size and volume diffusion variations caused by different hemoglobin/hematocrit ratios is depositing on the DBS card the precise volume of whole blood as a requirement of the pre-analytical step [40,41,42]. The approach displays perfect robustness and analytical stability of sample across 3 months at a wide range of storage temperatures [40]. Employment of the precise volume deposition allows storing the samples at both room temperature and in freezer (at −20 °C), with no significant impact on the final measurements (variation less than 20%) even for moderately stable substances and their metabolites [41]. The advantage of a precise volume deposed on the DBS for further analysis is also revealed in the good agreement of the DBS measurement results with those obtained from the liquid peripheral blood [43]. The validity and stability of measuring results are endorsed if precise volume deposing is combined with extraction by solvents fortified by internal heavy isotope-labeled standards [43,44]. Measurement of group B vitamins from the liquid whole blood and a precise volume of DBS using standards demonstrates an accuracy within less than 5% and a correlation at R^2^ = 0.96 [44].

## 4. Modern Analytical Methods for Dried Blood Spot Analysis

### 4.1. RNA/DNA Quantification

The use of DBS in virus screening and detection of nucleic acids (RNA, DNA) by quantitative polymerase chain reaction (qPCR) or reverse transcription polymerase chain reaction (RT-PCR) is of growing interest. These methods require a small sample volume (<20 µL) due to their ultrahigh sensitivity. However, it is important to note that the amount of material obtained from DBS samples is 1–2 log lower than those available in standard serum or plasma samples. Nucleic acids are stable on MC for an extended period of time [45] if samples are dried and stored away from moisture in accordance with the recommendations. PCR detection of the nucleic acids eluted from the DBS surface is primarily used in screening for viral diseases such as cytomegalovirus, herpes simplex virus [46], hepatitis B, hepatitis C [47] and HIV [48].

### 4.2. Protein/Peptide Detection

The use of DBS for the quantification of proteins and peptides is associated with certain limitations related to the relative complexity of their extraction from the MC surface. Nevertheless, it is possible to detect relatively abundant serum peptides or proteins/antibodies following-up the elution from DBS. The most widely used analytical methods for isolation of peptides and proteins are immunological assays, which identify analytes with high specificity and sensitivity. The immunoturbidimetric analysis of glycated hemoglobin for monitoring glycemic balance in patients with diabetes is one of the most explicit examples. The extraction and quantification of glycated hemoglobin from the surface of DBS correlates with that observed in standard immunoturbidimetric testing. In addition, this analyte remains stable for more than 15 days [49].

DBS technology is also well adapted for use with enzyme-linked immunosorbent assay (ELISA) reactions, allowing high selectivity, reproducibility and specificity of protein detection via antigen/antibody interactions. Specifically, it has been reported on repeated occasion that studies have achieved successful isolation of samples from MC surfaces, which were subsequently analyzed by ELISA for the presence of antibodies against Epstein–Barr virus [50], rubella virus [51], dengue virus [52], hepatitis C virus [53] and HIV virus [54].

In recent years, the combination of DBS and mass spectrometric analysis has become increasingly popular in biomedical research. Mass spectrometric platforms permit both surveying and targeted analysis of small molecules and proteins/peptides in biological samples of dried blood. Preliminary fractionation of a complex biological matrix using high-performance liquid chromatography mass spectrometry (HPLC/MS) allows for the quantitative measurement of peptides and proteins eluted from DBS [55]. Particularly, this approach has been adapted to measure ceruloplasmin for neonatal screening of Wilson’s disease [56] and to quantify C-peptide as a first line screening strategy for the monitoring of normal beta cell function [55].

Moreover, with support from a multiplex system, the mass spectrometry platform has the potential to quantify a huge number of protein analytes from the small volume of biomaterial. For example, using this approach, Chambers et al. [57] effectively quantified a panel of 40 serum proteins eluted from the DBS surface.

### 4.3. Metabolite Detection

High-resolution mass spectrometry (HRMS) has the potential to detect hundreds to thousands of metabolites in a single analysis and is readily used for a broad overview of the metabolome [58]. For this purpose, Velden M.G.M et al. [59] developed a chip based on nanospray ionization (nanoESI), which provides direct input of a biological sample. The primary advantages of this system include high sample throughput, no sample transfer, and the use of small volumes of biological material. Authors examined whether the nanoESI-HRMS chip is suitable for quantitative determination of metabolites important for the diagnosis of congenital metabolic disorders using DBS. The total time required for the detection of 21 analytes was 40 min. Diagnostically significant metabolites were quantified using nanoESI-HRMS-based chips and were found to correlate well with the results attained by the target liquid chromatography-tandem mass spectrometry.

Multiple reaction monitoring (MRM) is an LC-MS/MS approach that has been used to identify [6,6 2H2] glucose and [U-13C6] glucose enrichment using DBS without prior derivatization. The obtained results prove that DBS can contribute significantly to the study and diagnosis of complex metabolic diseases such as type 2 diabetes [60].

DBS technology is also applicable for dissociation-enhanced lanthanide fluorescent immunoassay. Zimmermann et al. used this technique to identify thyroglobulin in dried spots of whole blood for the analysis of thyroid function in children [61]. Moreover, sandwich immunoanalyzer flowmetrics based on Luminex xMAP technology can be adapted to work with DBS cards, as has been demonstrated in the detection of inflammatory markers [62].

The usage of DBS is not limited to clinical application solely. Advances in the development of materials for MCs with emphasis on improvements of long-term storage and easy transportation resulted in better extractability and higher recovery of low-molecular-weight compounds. This augmented the application of DBS to toxicology assay, doping control, and therapy monitoring. Odoardi S et al. used DBS to record narcotic substances and their metabolites (opiates, methadone, fentanyl and analogues, cocaine, amphetamines and amphetamine-like substances, ketamine, LSD) using ultra-performance liquid chromatography–tandem mass spectrometry (UHPLC-MS/MS). The limit of detection of the developed method was 0.05–1 ng/mL, and the limit of quantitation was 0.2–2 ng/mL. The method showed satisfied linearity for all substances, with determination coefficients exceeding 0.99. The developed technique was applied to authentic post-mortem blood samples [63].

Moretti M. et al. showed the possibility of quantifying cocaine, benzoylecgonine, ecgonine methyl ester and cocaethylene in post-mortem blood samples using DBS and the LC-MS/MS method. The estimated limits of detection and quantification were 1.0 and 5.0 ng/mL for cocaine and cocaethylene, and 0.5 and 2 ng/mL for ecgonine methyl ester and benzoylecgonine, respectively [41].

Some studies show that DBS can be used for forensic toxicology. S.S. Sadler’s team developed a method for the simultaneous determination of 11 illicit drugs using DBS in combination with UPLC-MS/MS technology [64]. The researchers also conducted an eight-month stability study at room temperature, 2–8 °C and −10 °C, and the best results were obtained at a temperature of −10 °C. Statistical analysis established an acceptable correlation between the attained results and those obtained by methods commonly used in the laboratory practice. This study determined that DBS could be an alternative or addition to the conventional analysis and sampling methods commonly used in forensic toxicology.

Currently, mass spectrometric methods allow the quantification of the analyte. In this way, using LC-MS/MS, ESI-MS/MS, IDES-MS/MS, FIA-ESI-MS/MS, the concentrations of biological substances such as adrenal steroids, amino acids profile, carnitine, creatine, acylcarnitines, etc. were detected [65]. However, to obtain reliable results of quantitative analysis, it is necessary to follow strict adherence to the rules. Thus, MCs that act as control or calibration MCs should be identical to MCs designed to collect patient biomaterial. If MCs of several types or from different manufacturers are used, additional comparison methods are required [37]. As mentioned earlier, the physical behavior of blotted whole blood is influenced by different parameters, such as hematocrit level, degree of hemolysis and anticoagulant type (if applicable). Currently, the hematocrit is recognized as the most meaningful parameter that affects the characteristics of the bloodstain, such as drying time, diffusion, uniformity, and also affects the reproducibility of the analysis. The hematocrit effect is more substantial when a sub-sample disk punch is analyzed, rather than the whole DBS specimen. Hence, method validation assay for DBS sample applications also needs to include examination of the impact of hematocrit variation on measurement and assay performance [37]. It is important to use an internal standard for the analysis of samples obtained using DBS. The use of a pre-processed internal standard MC ensures that both components of the internal standard and analytes are equally distributed on the carrier and will be extracted with equal efficiency. In addition, using manual extraction methods, an internal standard is added to the eluting reagent. In this case, the internal standard is extracted together with the target analyte. Another simple alternative is to introduce an internal standard directly into the sample being analyzed. Cross-contamination of samples is also one of the problems for analysis using DBS technology. The reasons for this may be different: for example, contact of cards during storage or the iterative use of punch cards. Therefore, it is recommended to perform the cleaning step or punch a blank card, especially for the analysis of compounds with low stability or for drugs monitoring [66]. To investigate the instrumentation carry-over effect, two injections of sequential blank DBS extracts should be performed after an injection of a sample with the upper limit of quantitation concentration. The response for the first and second blank matrix should not exceed 20% and 5%, respectively, of the mean response of the lower limit of detection of the analyte of interest [67]. Preparation of standards for quantitative analysis involves enrichment of whole blood with a set of commercial or proprietary calibration materials before validation. This may mean replacing a certain amount of the native plasma with artificial plasma containing a known concentration of the target analyte. The percentage of plasma-replaced cells must be minimized to prevent a solvent effect that creates a mismatch between spiked samples (calibrators) and patient samples.

Thus, the ability to adapt DBS technology to a myriad of clinical diagnostic platforms can contribute to the formation of analytical methods using a new generation of MC for a wide range of applications.

## 5. Practical Applications

Due to its many advantages, DBS technology is widely used for blood collection and storage in a variety of fields. The consequent analysis of biomaterial in DBS samples allows for the identification and quantification of a wide range of analytes. McDade et al. [68] described the possibility of identifying a minimum of 45 analytes, foremost, proteins from a single dried spot, which may prove critical for population studies. Among them, there were biomarkers characterizing the condition of endocrine, cardiovascular, reproductive and immune systems. High reproducibility of specificity and sensitivity in diagnostic using DBS and blood serum was demonstrated in 99%–100% of tested cases [26].

### 5.1. Newborn Screening

Newborn screening is one of the main achievements of public health, and is a comprehensive system that includes diagnosis, monitoring, treatment, and evaluation of newborns’ state reported after delivery [69]. It is required for the early detection of congenital metabolic, endocrine, hematological and other genetic diseases. In the last decade, thanks to the advent of tandem mass spectrometry, the sophisticated screening of newborns has become a mandatory health strategy in most developed countries, which provides timely interventions that lead to a significant reduction in morbidity, mortality and disability [70]. Newborn screening using mass spectrometry, which allows the simultaneous detection of more than 30 different metabolic disorders in one DBS sample [71]. Using MS/MS to screen inborn disturbances of metabolism provides benefits such as high analytical sensitivity, selectivity and accuracy, with the ability to measure multiple analytes in a single analysis with relatively low noise [72]. Most of the limitations of this method are related to the principle of blood sampling, chromatographic and volumetric effects, analyte stability and hematocrit [15]. However, these limitations can be resolved through strict adherence to common standard operating procedures [39]. Screening is performed by analysis of blood taken from the heel of the newborn using a special filter paper. A sample is collected by a healthcare professional—usually in the maternity ward—during the first 24–48 h of life.

For the very first time, screening of newborns for phenylketonuria (PKU) was proposed by Robert Guthrie. R., who developed a bacterial inhibition test for the semi-quantitative analysis of phenylalanine. An early diagnosis of PKU and a subsequent low-phenylalanine diet prevents the development of severe mental retardation. In addition, he introduced the Guthrie special filter paper as a means of transporting blood, which is still used [73]. DBS is also used to detect lysosomal diseases in newborns by measuring lysosomal enzymatic activity [74]

Céspedes N. et al. demonstrated a method for the simultaneous assessment of 57 analytes (amino acids, acylcarnitines and succinylacetone) associated with more than 40 congenital metabolic errors using tandem mass spectrometry (MS/MS). The analysis of DBS samples enriched with analyte and subsequently mixed with labeled internal standards (isotopic dilution method) showed a linear relationship between the data at a wide range of concentrations, with high sensitivity [75].

Congenital metabolic errors are a phenotypically and genetically heterogeneous group of a large number of rare disorders that are caused by genetic defects. These defects may alter the ability to extract necessary energy from nutrients, leading to metabolic disorders and/or the accumulation of toxic intermediate metabolites. X-linked adrenoleukodystrophy (X-ALD) is a peroxisomal disorder caused by mutations in the ABCD1 gene and affects approximately 1 in 42,000 men [76,77]. In addition, 1 out of 28,000 women carry heterozygous mutations in ABCD1 [77]. Newborn screening for X-ALD is based on recording an elevated level of a very-long-chain fatty acid derivative of lysophosphatidylcholine (C26: 0-LPC) in DBS samples. In the first step, MS/MS analysis is used to measure C26: 0-LPC. Then, when samples with increased C26: 0-LPC are detected, MS/MS is used in combination with high-performance liquid chromatography to measure C26: 0-LPC in the same spot of dry blood [78].

Severe combined immunodeficiency (SCID) is a hereditary disorder of the immune system caused by a spectrum of genetic defects leading to cellular and humoral immunodeficiency [79]. It has been shown that T-cell receptor excision circles (TRECs), which are a SCID marker, can be detected using DBS [80]. Kappa-deleting recombination excision circle (KREC) is present in 30% of Igκ^+^ and almost all Igλ^+^ mature naive B lymphocytes [81]. KREC DNA can also be a surrogate marker for mature naive B lymphocytes and can also be used to evaluate their proliferative history [82]. The combined TREC/KREC assay identifies individuals with various forms of primary immunodeficiency diseases that might be omitted by the TREC assay solely, including adenosine deaminase enzyme deficiency, some cases of Nijmegen syndrome, and X-linked or autosomal recessive agammaglobulinemia [83]. Barbaro M. et al. simultaneously recognized TREC and KREC using quantitative PCR for DNA isolated by DBS [84]. A highly promising diagnostic level was estimated to identify newborns with milder and more reversible T and/or B-cell lymphopenia.

Ludwig Czibere et al. developed a test using the DBS nucleic acid isolation protocol and quantitative (q) PCR to screen for the homozygous deletion of exon 7 of the motor neuron 1 (SMN1) gene survival, which is responsible for the development of spinal muscular atrophy in newborns in 95% of cases. The research group tested 213,279 samples and, eventually, identified the disease in a timely manner and determined treatment options before symptom appearance. [85]

### 5.2. Serodiagnosis

Application of DBS technology for the detection of antibodies as markers of infectious diseases [86], including viral pathogens (hepatitis, rubella, measles, etc.) [50,51], bacterial infections (tetanus, leptospirosis, brucellosis, leishmaniasis) and helminths (filariasis) [87] has been extensively described. Moreover, the expediency of using this inexpensive and effective approach for HIV surveillance [19], monitoring of hormonal status [88], and food intolerances [3] has also been demonstrated. The most striking advantage of DBS is the requirement for a very small blood volume to be used in combination with highly sensitive analytical methods, such as MRM-MS, or in combination with fractionation and high-resolution LC-MS/MS platforms.

Analysis of multiple metabolites allows the diagnosis and treatment of human pathological conditions. For example, screening for congenital adrenal hyperplasia (CAH) is accomplished via detection of 17-hydroxyprogesterone (17-OHP). In 1977, an immunoassay approach for the quantitative determination of 17-OHP in dried blood samples for newborns was developed [89]. Unfortunately, the positive prognostic value of screening for congenital adrenal hyperplasia populations by immunoassay remains below 1%, due to delayed expression of 11-hydroxylase in premature infants. Causes for this condition include stress at birth, impaired renal function, the buildup of 17-OHP metabolites, and cross-reactivity between antibodies and steroids, in particular 17-hydroxypregnenolone (17-Ophrah). The molecular weight of 17-OHP is 330 Dalton (Da), which is two Dalton units less than that of the competitive compound, 17-Ophrah (332 Da), and both compounds can be distinguished via mass spectrometry using any detection mode (Q1 scan, product scan, or selected reaction monitoring) [89].

### 5.3. Bioenvironmental Monitoring

Persistent organic pollutants, such as polychlorinated biphenyls and organochlorine pesticides, are bio-accumulative, and toxic substances and spread over the atmosphere across long distances [90]. Numerous studies have shown that persistent organic pollutants have an adverse effect on human health, including impaired endocrine, reproductive and immunological systems and may cross the trans-placental barrier [91,92,93,94]. High concentrations of hexachlorobenzene and polychlorinated biphenyl, reported in women’s plasma samples, were associated with a risk of type 2 diabetes [95]. An increased risk of childhood obesity has been associated with the prenatal effects of polychlorinated biphenyl and dichlorodiphenyldichloroethylene [96]. Fetuses, newborns and infants are more sensitive to the effects of environmental chemicals than adult subjects [97].

In the study conducted by Ma W.L et al. [98], polychlorinated biphenyls and organochlorine pesticides were measured using high-resolution gas chromatography mass spectrometry (GC-HRMS) in DBS samples from newborns. The measured concentrations of polychlorinated biphenyls and dichlorodiphenyldichloroethylene in the peripheral blood of newborns were comparable with the concentrations obtained in the analysis of the umbilical blood, indicating a trans-placental transfer of these compounds from mother to fetus.

The most commonly encountered polybrominated diphenyl ethers identified in DBS samples using GC-HRMS are polybrominated diphenyl ethers-47, polybrominated diphenyl ethers-99 and polybrominated diphenyl ethers-100 [99]. The presence of polybrominated diphenyl ethers in the peripheral blood of newborns confirms the possibility of transporting these compounds across the placental barrier. Dr. Ghassabian et al. showed that increased prenatal exposure to perfluorooctane sulfonic acid or perfluorooctanoic acid can contribute to the occurrence of behavioral disorders in children [100]. The method of mass spectrometry with high-resolution gas chromatography supported with ultrasound extraction has been successfully applied to the analysis of persistent organic pollutants in DBS samples.

Thus, DBS is a valuable resource for studying of the adverse impact of environmental chemicals on the newborns and fetus growth. A significant improvement in analytical technologies will probably allow the analysis of target environmental chemicals from one dry spot. However, a major limitation that requires further study is the determination of the exact volume of whole blood in a DBS sample.

## 6. Limitation of DBS in Biomedical Research

Despite the fact that DBS technology has achieved great success in the field of bioanalysis, certain questions remain to be answered. The DBS technology uses capillary blood from volunteers in clinical studies; however, the concentration of analytes in capillary blood may differ from that in venous blood. For example, it was reported that paracetamol has a significantly higher concentration in capillary relative to venous blood [101]. In studies examining the long-term stability of MC, a change in concentration over time was reported. Specifically, carnitine levels were found to increase by 7.6 % per year during the first five years, followed by a 1.4 % decrease each subsequent year [102].

Precautions are also required when applying DBS to metabolic profiling techniques. It has been reported that specific metabolites, including L-lysine, iminodiacetic acid, DL-Treo-beta-hydroxyaspartic acid, citric acid, adipamide and adenosine-5-monophosphate are absent in DBS eluates, yet are present in blood and plasma [103]. Moreover, when analyzing analytes extracted via DBS, the potential interaction between the analyte and MC components can lead to suppression of ionization in the MS source, as well as changes in chromatographic mobility and peak sharpness. Therefore, when using DBS technology, it is necessary to take into account the stability of the analyte on the cards during drying and storage, the effect of hematocrit, the homogeneity of the spot and the efficiency of eluting the analyte.

## 7. Biobank

Medical and biological studies rely on the analysis of clinical information and data, following the collection of biological samples [104]. Thus, accessibility to standard clinical samples often impedes completion of scientific studies. Moreover, the collection of biological samples required for research over extended periods (an average of 6 months) is complex and requires expensive organization and supporting equipment. It also requires established contact with the profile clinic, and their agreement to participate in sample collection, to maintain storage conditions and transport biological samples in accordance with the recommendations of the customer. The necessity to create targeted banks of biological materials has been discussed in the scientific community [105]. For example, the Danish National Biobank for neonatal screening includes more than two million DBS for all Danes born since 1982 [1]. Bioresource collections are the constituents of scientific organizations designed to provide access for Russian and foreign scientists to biological specimens used in fundamental and applied research.

DBS membrane carriers remaining after the screening can serve as a basis for the conceptualization of a Biobank of DBS maps, which would contain cell elements for genomic analysis, plasma proteins and metabolites for metabolomic and proteomic profiling, and cell homogenates and culture fluids obtained during the cultivation of cell cultures. Thus, MCs serve as a source of biomaterial for repeated studies in cases of dubious test results.

## 8. Conclusions

Despite specific limitations, the DBS sampling method is undoubtedly the most ethical and inexpensive method of collecting, delivering and storing the biomaterial. This new approach to blood collection is widely used in clinical and pharmaceutical laboratories and offers a number of advantages: (1) it is a simple, minimally invasive sampling process; (2) it provides a convenient means to store and transfer biomaterial, due to the lack of requirements for the use of deep freeze chambers; (3) biomaterial can be easily collected by the patient on their own; (4) it provides the possibility of long-distance transportation with minimal costs [9]; (5) it reduces the risk of bloodborne transmission [20]; and (6) in contrast to traditional methods for blood samples collection, it requires a smaller volume (less than 100 µL) of sample, which greatly simplifies the procedure.

Herein, we have demonstrated the prospect of using this technology to assess the metabolomic profile, clinical assessment, diagnosis of communicable diseases, etc. Improved DNA detection methods provide a mechanism for multicomponent molecular testing of single samples and the possibility of obtaining genetic information when testing dried spots. The question of who should have the access to dried blood spot banks, and how to maximize the use of this unique resource in the best way, is actively debated. In addition, it is necessary to develop a clear, science-based, legally-supported and coordinated policy between the necessary ministries and departments concerned, for the integration of DBS technology in population studies of human health.

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
