# Peer review of "Dried Blood Spot in Laboratory: Directions and Prospects"

_diagnostics, 2020, doi:10.3390/diagnostics10040248_

Round 1

Reviewer 1 Report

The manuscript refers to a review that illustrates the information available through the DBS method, which may be useful for biomedical researchers.

It describes some modern analytical methods for DBS analysis and explores some practical application of DBS samples.

The review presented in this manuscript are interesting and well organized.

However, the review cannot be considered complete.

A] it is necessary to specify how the literature review was conducted (sources, search protocol, strategy to identify the most relevant evidence...).

B] The discussion does not address some use of DBS, more or less widespread, such as therapeutic drug monitoring, pharmacokinetics, toxicokinetics and toxicological analysis. In fact, DBS is gaining interest in the forensic field in the last years, and several methods for the identification and quantification of drugs of abuse and psychoactive substances have been developed, even in post-mortem samples. Various examples: 

  1. Moretti, M et all A liquid chromatography-tandem mass spectrometry method for the determination of cocaine and metabolites in blood and in dried blood spots collected from postmortem samples and evaluation of the stability over a 3-month period. Drug Test Anal 2018, 10, 1430-1437, doi:10.1002/dta.2399.
  2. Odoardi, S.; Anzillotti, L.; Strano-Rossi, S. Simplifying sample pretreatment: application of dried blood spot (DBS) method to blood samples, including postmortem, for UHPLC-MS/MS analysis of drugs of abuse. Forensic Sci Int 2014, 243, 61-67, doi:10.1016/j.forsciint.2014.04.015.
  3. Chepyala, D.; Tsai, I.L.; Liao, H.W.; Chen, G.Y.; Chao, H.C.; Kuo, C.H. Sensitive screening of abused drugs in dried blood samples using ultra-high-performance liquid chromatography-ion booster-quadrupole time-of-flight mass spectrometry. J Chromatogr A 2017, 1491, 57-66, doi:10.1016/j.chroma.2017.02.037.
  4. Saussereau, E.; Lacroix, C.; Gaulier, J.M.; Goulle, J.P. On-line liquid chromatography/tandem mass spectrometry simultaneous determination of opiates, cocainics and amphetamines in dried blood spots. J Chromatogr B Analyt Technol Biomed Life Sci 2012, 885-886, 1-7, doi:10.1016/j.jchromb.2011.11.035.
  5. Sadler Simoes, S.; Castanera Ajenjo, A.; Dias, M.J. Dried blood spots combined to an UPLC-MS/MS method for the simultaneous determination of drugs of abuse in forensic toxicology. J Pharm Biomed Anal 2018, 147, 634-644, doi:10.1016/j.jpba.2017.02.046.
  6. Kyriakou, C.; Marchei, E.; Scaravelli, G.; Garcia-Algar, O.; Supervia, A.; Graziano, S. Identification and quantification of psychoactive drugs in whole blood using dried blood spot (DBS) by ultra-performance liquid chromatography tandem mass spectrometry. J Pharm Biomed Anal 2016, 128, 53-60, doi:10.1016/j.jpba.2016.05.011.
  7. Ambach, L.; Hernandez Redondo, A.; Konig, S.; Weinmann, W. Rapid and simple LC-MS/MS screening of 64 novel psychoactive substances using dried blood spots. Drug Test Anal 2014, 6, 367-375, doi:10.1002/dta.1505.
  8. Deglon, J.; Lauer, E.; Thomas, A.; Mangin, P.; Staub, C. Use of the dried blood spot sampling process coupled with fast gas chromatography and negative-ion chemical ionization tandem mass spectrometry: application to fluoxetine, norfluoxetine, reboxetine, and paroxetine analysis. Anal Bioanal Chem 2010, 396, 2523-2532, doi:10.1007/s00216-009-3412-6.
  9. Deglon, J.; Thomas, A.; Cataldo, A.; Mangin, P.; Staub, C. On-line desorption of dried blood spot: A novel approach for the direct LC/MS analysis of micro-whole blood samples. J Pharm Biomed Anal 2009, 49, 1034-1039, doi:10.1016/j.jpba.2009.02.001.

It is necessary at least to mention these aspects in the manuscript, and the references should be implemented.

Otherwise, it is necessary to specify in the title and introduction that the review is limited to some clinical areas.

C] At line 156/169 and also at 236/243 authors talk about problems that could influence the diffusion of blood on DBS cards and the size of the spot (such as hematocrit).

In order to overcome these issues, some authors tried to analyze the whole blood spot deposed on the paper substrate after deposing an exact amount of blood (for example see   Moretti, et all. Determination of benzodiazepines in blood and in dried blood spots (DBS) collected from postmortem samples and evaluation of the stability over a three-month period. Drug Test Anal 2019, doi:10.1002/dta.2653.   and Moretti, M.; et all. Determination of benzodiazepines in blood and in dried blood spots (DBS) collected from postmortem samples and evaluation of the stability over a three-month period. Drug Test Anal 2019, doi:10.1002/dta.2653).

D] English improvements is required. Pay attention to grammatical errors and typos. Some examples:

  • Line 11: "becomes absorption" should be "becomes absorbed".
  • in line  15 "...including the requirement for small volumes of biomaterials, transportation and storage of samples do not require special conditions, improved stability of analytes and reduced risk of infection resulting from contaminated samples". A relative pronoun is missing
  • lines 37-39: check the sentence (term “disease” is repeated 3 times)
  • line 38: remove the first comma
  • lines 45/47: the sentence is not clear
  • lines 67-68: more descriptions are needed about references 12, 13
  • line 72: check the sentence: “From the point of view of clinical practice, it is important to note that capillary blood provides sufficient sample size for processing by DBS” .
  • lines 78-83: the sentence is too long, improve readability
  • lines 112-113: check the sentence
  • line 122: check the sentence.
  • line 129: “..a certain level of large DNA loci were.” à Was
  • line 138-139:  check this sentence "Once such method for employed for the modification of existing membrane materials "
  • line 142: “allowing for preservation of sample composition
  • line 151: “quantification. [34] It….” move the point.
  • line 161: “MV” à MC
  • line 192: “…allowing for high selectivity,…”
  • line 198: “allow for both panoramic..”
  • lines 209-211: “…potential to detect hundreds to thousands of metabolite …”… “. allows for the simultaneous detection of thousands of metabolite masses..” repeated concepts.
  • line 237: “…by different parameters such as; haematocrit level, degree…”. Use “:”
  • line 288: “..enzymatic activity[65].”. space
  • line 299: “…allows for the diagnosis..”
  • line 300: “…is one such example…”
  • line 320 instead of “in this study [78]…”, specify “In the study performed by Ma W.L at al. [78]…”
  • Lines 326-330: repeated paragraph!
  • Line 388 “…hemocontact infection…”. Not sure about that
  • Line 395: “…dried Blood Spot..” capital letter

E) Some parts are a little repetitive (for example, the authors repeated several times the advantages of DBS or the issues related to hematocrit effect).

Author Response

Please find attached a revision of the original manuscript. We have taken into account the reviewers’ comments and have addressed these in the revision. Corrections and additions of references are highlighted in green in this text.

Reviewer 1
The manuscript refers to a review that illustrates the information available through the DBS method, which may be useful for biomedical researchers.
It describes some modern analytical methods for DBS analysis and explores some practical application of DBS samples.
The review presented in this manuscript are interesting and well organized.
However, the review cannot be considered complete.
Question 1
A] it is necessary to specify how the literature review was conducted (sources, search protocol, strategy to identify the most relevant evidence...).
Answer 1
We have added a new section “Search strategy”, in which we described a literature search strategy (pp. 2-3, lines 94-106).
Question 2
B] The discussion does not address some use of DBS, more or less widespread, such as therapeutic drug monitoring, pharmacokinetics, toxicokinetics and toxicological analysis. In fact, DBS is gaining interest in the forensic field in the last years, and several methods for the identification and quantification of drugs of abuse and psychoactive substances have been developed, even in post-mortem samples. Various examples:
1. Moretti, M et all A liquid chromatography-tandem mass spectrometry method for the determination of cocaine and metabolites in blood and in dried blood spots collected from postmortem samples and evaluation of the stability over a 3-month period. Drug Test Anal 2018, 10, 1430-1437, doi:10.1002/dta.2399.
2. Odoardi, S.; Anzillotti, L.; Strano-Rossi, S. Simplifying sample pretreatment: application of dried blood spot (DBS) method to blood samples, including postmortem, for UHPLC-MS/MS analysis of drugs of abuse. Forensic Sci Int 2014, 243, 61-67, doi:10.1016/j.forsciint.2014.04.015.
3. Chepyala, D.; Tsai, I.L.; Liao, H.W.; Chen, G.Y.; Chao, H.C.; Kuo, C.H. Sensitive screening of abused drugs in dried blood samples using ultra-high-performance liquid chromatography-ion booster-quadrupole time-of-flight mass spectrometry. J Chromatogr A 2017, 1491, 57-66, doi:10.1016/j.chroma.2017.02.037.
4. Saussereau, E.; Lacroix, C.; Gaulier, J.M.; Goulle, J.P. On-line liquid chromatography/tandem mass spectrometry simultaneous determination of opiates, cocainics and amphetamines in dried blood spots. J Chromatogr B Analyt Technol Biomed Life Sci 2012, 885-886, 1-7, doi:10.1016/j.jchromb.2011.11.035.
5. Sadler Simoes, S.; Castanera Ajenjo, A.; Dias, M.J. Dried blood spots combined to an UPLC-MS/MS method for the simultaneous determination of drugs of abuse in forensic toxicology. J Pharm Biomed Anal 2018, 147, 634-644, doi:10.1016/j.jpba.2017.02.046.
6. Kyriakou, C.; Marchei, E.; Scaravelli, G.; Garcia-Algar, O.; Supervia, A.; Graziano, S. Identification and quantification of psychoactive drugs in whole blood using dried blood spot (DBS) by ultra-performance liquid chromatography tandem mass spectrometry. J Pharm Biomed Anal 2016, 128, 53-60, doi:10.1016/j.jpba.2016.05.011.
7. Ambach, L.; Hernandez Redondo, A.; Konig, S.; Weinmann, W. Rapid and simple LC-MS/MS screening of 64 novel psychoactive substances using dried blood spots. Drug Test Anal 2014, 6, 367-375, doi:10.1002/dta.1505.
8. Deglon, J.; Lauer, E.; Thomas, A.; Mangin, P.; Staub, C. Use of the dried blood spot sampling process coupled with fast gas chromatography and negative-ion chemical ionization tandem mass spectrometry: application to fluoxetine, norfluoxetine, reboxetine, and paroxetine analysis. Anal Bioanal Chem 2010, 396, 2523-2532, doi:10.1007/s00216-009-3412-6.
9. Deglon, J.; Thomas, A.; Cataldo, A.; Mangin, P.; Staub, C. On-line desorption of dried blood spot: A novel approach for the direct LC/MS analysis of micro-whole blood samples. J Pharm Biomed Anal 2009, 49, 1034-1039, doi:10.1016/j.jpba.2009.02.001.
It is necessary at least to mention these aspects in the manuscript, and the references should be implemented.
Otherwise, it is necessary to specify in the title and introduction that the review is limited to some clinical areas.
Answer 2
We have included a part of the recommended sources in the new subsection “4.3. Metabolite detection” (p. 5, lines 232-311) encompassing various application of DBS in toxicology, metabolomics and observation of drugs of abuse.
Question 3
C] At line 156/169 and also at 236/243 authors talk about problems that could influence the diffusion of blood on DBS cards and the size of the spot (such as hematocrit).
In order to overcome these issues, some authors tried to analyze the whole blood spot deposed on the paper substrate after deposing an exact amount of blood (for example see Moretti, et all. Determination of benzodiazepines in blood and in dried blood spots (DBS) collected from postmortem samples and evaluation of the stability over a three-month period. Drug Test Anal 2019, doi:10.1002/dta.2653. and Moretti, M.; et all. Determination of benzodiazepines in blood and in dried blood spots (DBS) collected from postmortem samples and evaluation of the stability over a three-month period. Drug Test Anal 2019, doi:10.1002/dta.2653).
Answer 3
We analyzed the recommended sources and added in the revised paper the relevant information about the strategy of accurate quantitation (p. 4, lines 181-193).
Question 4
D] English improvements is required. Pay attention to grammatical errors and typos. Some examples:
• Line 11: "becomes absorption" should be "becomes absorbed".
• in line 15 "...including the requirement for small volumes of biomaterials, transportation and storage of samples do not require special conditions, improved stability of analytes and reduced risk of infection resulting from contaminated samples". A relative pronoun is missing
• lines 37-39: check the sentence (term “disease” is repeated 3 times) line 38: remove the first comma
• lines 45/47: the sentence is not clear
• lines 67-68: more descriptions are needed about references 12, 13
• line 72: check the sentence: “From the point of view of clinical practice, it is important to note that capillary blood provides sufficient sample size for processing by DBS”.
• lines 78-83: the sentence is too long, improve readability
• lines 112-113: check the sentence
• line 122: check the sentence.
• line 129: “..a certain level of large DNA loci were.” à Was
• line 138-139: check this sentence "Once such method for employed for the modification of existing membrane materials "
• line 142: “allowing for preservation of sample composition”
• line 151: “quantification. [34] It….” move the point.
• line 161: “MV” à MC
• line 192: “…allowing for high selectivity,…”
• line 198: “allow for both panoramic..”
• lines 209-211: “…potential to detect hundreds to thousands of metabolite …”… “. allows for the simultaneous detection of thousands of metabolite masses..” repeated concepts.
• line 237: “…by different parameters such as; haematocrit level, degree…”. Use “:”
• line 288: “..enzymatic activity[65].”. space
• line 299: “…allows for the diagnosis..”
• line 300: “…is one such example…”
• line 320 instead of “in this study [78]…”, specify “In the study performed by Ma W.L at al. [78]…”
• Lines 326-330: repeated paragraph!
• Line 388 “…hemocontact infection…”. Not sure about that
• Line 395: “…dried Blood Spot..” capital letter
Answer 4
We totally agree with the claim of English language improvement. We followed throughout the paper and attempted to fix errors and typos including those indicated by the Reviewer. All changes are highlighted in green.
Question 5
E) Some parts are a little repetitive (for example, the authors repeated several times the advantages of DBS or the issues related to hematocrit effect).
Answer 5
We tried to exclude textual repetitions.

Yours faithfully, authors

Reviewer 2 Report

This review consists of an update of the advantages, limitations and challenges presented by DBS technology for applications in biomedicine as in biomedical research.

MAJOR COMMENTS

The term "sample" is confused with "specimen" throughout the document. Please, change "sample" to "specimen" where applicable (eg Lines 86, 119, 241 or 265).

MINOR COMMENTS

Line 93. The acronym "MC" was previously defined in Line 24. It is not necessary to define it again.

Line 112. A dot is necessary after "drying".

Line 161. What is "MV"? I suppose it is a mistake. If not, please define.

Line 232. Include "acylcarnitines"

Line 252. In practice, it is not common done or recommended, except for DNA analysis. Please specify.

Line 280. Change "filter card" to "special filter paper" or "specimen collection device". This terms are common used for newborn screening but not "filter card".

Lines 287-288. In the last decade, many advances have been made in the detection of genetic diseases through newborn screening, not only for lysosomal disorders. I suggest that you include a paragraph explaining the new diseases and technologies that are being applied (eg. amino acids and acylcarnitines for expanded newborn screening, C26:0 lysophosphatidylcholine for X-ALD, TRECs and KREKs analysis for SCID, DNA anaylisis for cystic fibrosis or spinal medular atrophy).

Author Response

Please find attached a revision of the original manuscript. We have taken into account the reviewers’ comments and have addressed these in the revision. Corrections and additions of references are highlighted in green in this text.

This review consists of an update of the advantages, limitations and challenges presented by DBS technology for applications in biomedicine as in biomedical research.
Question 1
MAJOR COMMENTS
The term "sample" is confused with "specimen" throughout the document. Please, change "sample" to "specimen" where applicable (eg Lines 86, 119, 241 or 265).
Answer 1
We followed the recommendation and corrected in the indicated places "sample" to "specimen".
Question 2
MINOR COMMENTS
Line 93. The acronym "MC" was previously defined in Line 24. It is not necessary to define it again.
Line 112. A dot is necessary after "drying".
Line 161. What is "MV"? I suppose it is a mistake. If not, please define.
Line 232. Include "acylcarnitines"
Line 252. In practice, it is not common done or recommended, except for DNA analysis. Please specify.
Line 280. Change "filter card" to "special filter paper" or "specimen collection device". These terms are common used for newborn screening but not "filter card".
Answer 2
We appreciate the Reviewer for the corrections and completely agree. We fixed all the correction throughout the paper and indicated them in green color.
2
Question 3
Lines 287-288. In the last decade, many advances have been made in the detection of genetic diseases through newborn screening, not only for lysosomal disorders. I suggest that you include a paragraph explaining the new diseases and technologies that are being applied (eg. amino acids and acylcarnitines for expanded newborn screening, C26:0 lysophosphatidylcholine for X-ALD, TRECs and KREKs analysis for SCID, DNA anaylisis for cystic fibrosis or spinal medular atrophy).
Answer 3
We thank the Reviewer for this valuable recommendation because the lack of information regarding the new technologies in newborn screening may reduce the impression of DBS applicability. Therefor we have included recommendations in section 5.1. Newborn screening (pp. 7-8, lines 324-326; lines 328-336; lines 344-379).

Yours faithfully, authors

Round 2

Reviewer 1 Report

I thank the authors for the additional work they have done. Revisions are satisfactory. 

The manuscript is now complete and well written.

I recommend this paper for publication. 

Author Response

Dear reviewers,
The team of authors is grateful for your constructive recommendations and comments. This greatly improved the quality of our review article.

Yours faithfully, authors